| 1  | A Generalization of the TRIX trophic index to the Adriatic Sea basin                                                                              |
|----|---------------------------------------------------------------------------------------------------------------------------------------------------|
| 2  | Emanuela Fiori <sup>1, 2</sup> , Marco Zavatarelli <sup>1, 2</sup> , Nadia Pinardi <sup>1, 2</sup> , Cristina Mazziotti <sup>3</sup> , Carla Rita |
| 3  | Ferrari <sup>3</sup>                                                                                                                              |
| 4  |                                                                                                                                                   |
| 5  | <sup>1</sup> Dipartimento di Fisica e Astronomia (DIFA) University of Bologna, Viale Berti Pichat,                                                |
| 6  | 6/2, 40127 Bologna, Italy                                                                                                                         |
| 7  | <sup>2</sup> Consorzio Nazionale Interuniversitario per le Scienze dal Mare, Piazzale Flaminio 9,                                                 |
| 8  | 00195 Roma, Italy                                                                                                                                 |
| 9  | <sup>3</sup> Agenzia Regionale per la prevenzione, l'ambiente e l'energia dell'Emilia-Romagna,                                                    |
| 10 | Struttura Oceanografica Daphne, Viale Vespucci, 2, 47042 Cesenatico (FC), Italy                                                                   |
| 11 |                                                                                                                                                   |
| 12 | Corresponding author: Emanuela Fiori                                                                                                              |
| 13 | Telephone: 0544-937322                                                                                                                            |
| 14 | Fax: 0544-937345                                                                                                                                  |
| 15 | e-mail: <u>e.fiori@sincem.unibo.it</u>                                                                                                            |
| 16 |                                                                                                                                                   |
| 17 |                                                                                                                                                   |
| 18 |                                                                                                                                                   |
| 19 |                                                                                                                                                   |
| 20 |                                                                                                                                                   |
| 21 |                                                                                                                                                   |
| 22 |                                                                                                                                                   |
| 23 |                                                                                                                                                   |

2

| 24 | Abstract                                                                                         |
|----|--------------------------------------------------------------------------------------------------|
| 25 | The Marine Strategy Framework Directive is pushing for new methodological approaches in          |
| 26 | order to protect the marine environment more effectively. The trophic index TRIX was             |
| 27 | developed by Vollenweider in 1998 for the coastal area of Emilia-Romagna (northern               |
| 28 | Adriatic Sea), and was exploited by Italian legislation to characterize the trophic state of     |
| 29 | coastal waters. In order to implement TRIX in different areas and for different time periods,    |
| 30 | we developed a methodology for the generalization of the index changing the scaling              |
| 31 | parameters.                                                                                      |
| 32 | We compared the TRIX index calculated from in situ data ("in situ TRIX") with the                |
| 33 | corresponding index simulated with a coupled physics and biogeochemical numerical model          |
| 34 | ("model TRIX") implemented in the overall Adriatic Sea. The comparison between in situ           |
| 35 | and simulated data was carried out for a data time series in the Emilia-Romagna coastal strip.   |
| 36 | This study demonstrates the compatibility of the model with the in situ TRIX and the             |
| 37 | necessity to have time series longer than 10 years to evaluate properly the scaling parameters.  |
| 38 | The model TRIX is finally calculated for the whole Adriatic Sea showing trophic index            |
| 39 | differences across the Adriatic coastal areas.                                                   |
| 40 |                                                                                                  |
| 41 | 1. Introduction                                                                                  |
| 42 | Marine habitats are subject to increasing pressures (as nutrients discharges, eutrophication)    |
| 43 | due to agriculture, industry, tourism, fishing, and aquaculture. The eutrophication of coastal   |
| 44 | waters is considered to be one of the greatest threats to the health of marine ecosystems. It is |
| 45 | described as a change in the marine food web connected to the seawater enrichment by             |
| 46 | nutrients, which can modify the carbon pathways and excessive oxygen consumption                 |
| 47 | (Ferreira et al., 2011; Vollenweider et al., 1992).                                              |
| 48 | In response to these pressures, the Marine Strategy Framework Directive (MSFD,                   |

2008/56/EC) explicitly considers eutrophication descriptors as key to determining the Good

| 50 | Environmental Status (GES) of European coastal waters. The MSFD underlines the need to          |  |  |  |  |
|----|-------------------------------------------------------------------------------------------------|--|--|--|--|
| 51 | implement an ecosystem-based approach to determine all the pressures affecting the marine       |  |  |  |  |
| 52 | environment relatively to the GES. Indicators therefore need to be developed to qualitatively   |  |  |  |  |
| 53 | and quantitatively assess the quality of the marine environment. Marine ecosystems also         |  |  |  |  |
| 54 | present high levels of complexity; hence indicators are needed to support monitoring            |  |  |  |  |
| 55 | programs and reduce complexity for early warning systems.                                       |  |  |  |  |
| 56 | Eutrophication assessment indicators should use multivariate water column state variables,      |  |  |  |  |
| 57 | integrating physical-chemical and biological variables. TRIX is an eutrophication index,        |  |  |  |  |
| 58 | proposed by Vollenweider et al. (1998) in order to characterize the trophic state of marine     |  |  |  |  |
| 59 | waters along the Emilia-Romagna coastal region (North Western Adriatic Sea). TRIX is            |  |  |  |  |
| 60 | defined by four state variables, which are strongly correlated with primary production:         |  |  |  |  |
| 61 | chlorophyll-a, oxygen, dissolved inorganic nitrogen and total phosphorous. The TRIX index       |  |  |  |  |
| 62 | was integrated into Italian law in order to monitor the status of the coastal marine            |  |  |  |  |
| 63 | environment (D.L. 260/2010 table 4.3.2/c).                                                      |  |  |  |  |
| 64 | TRIX covers a wide range of trophic conditions from oligotrophy to eutrophy and it has been     |  |  |  |  |
| 65 | applied to coastal marine waters in several European seas: the Adriatic Sea and the             |  |  |  |  |
| 66 | Tyrrhenian Sea (Giovanardi and Vollenweider, 2004), the Black Sea (Kovalova and                 |  |  |  |  |
| 67 | Medinets, 2012; Baytut, 2010; Dayatlov et al., 2010; Medinets et al. 2010; Moncheva and         |  |  |  |  |
| 68 | Doncheva 2000; Moncheva et al., 2002; Zaika, 2003), the eastern Mediterranean Sea (Tugrul       |  |  |  |  |
| 69 | et al., 2011), the Aegean Sea (Yucel-Gier et al., 2011), the Marmara Sea (Balkis et al., 2012), |  |  |  |  |
| 70 | the Caspian Sea (Shahrban and Etemad-Shahidi, 2010), the Mar Menor Lagoon (Salas et al.,        |  |  |  |  |
| 71 | 2008), the Persian Gulf (Zoriasatein et al., 2013), and the Gulf of Finland (Vaschetta et al.,  |  |  |  |  |
| 72 | 2004).                                                                                          |  |  |  |  |
| 73 | However, the general methodology for constructing TRIX for the different areas has not been     |  |  |  |  |
| 74 | clarified. In order to apply TRIX to different areas a precise evaluation of the scaling        |  |  |  |  |
| 75 | parameters is required.                                                                         |  |  |  |  |

| 76  | In this paper we review the methodology and explain how to adapt TRIX to different coastal                        |
|-----|-------------------------------------------------------------------------------------------------------------------|
| 77  | and open ocean areas. The specific objectives of our work are: (1) to develop a generic TRIX                      |
| 78  | index equation for the coastal and open ocean areas in the entire Adriatic Sea; (2) to adapt the                  |
| 79  | TRIX to numerical ecosystem model simulation data; (3) to validate the "model TRIX" with                          |
| 80  | in situ data for a long time period. The final results of this paper present a new TRIX index                     |
| 81  | formulation that could be used to assess the GES of coastal and marine waters in terms of                         |
| 82  | eutrophication status.                                                                                            |
| 83  | Section 2 describes the TRIX equation and its calibration parameters. Section 3 illustrates the                   |
| 84  | in situ and simulation model data used for the evaluation of TRIX and its calibration. Section                    |
| 85  | 4 compares the "in situ TRIX" and "model TRIX", and the sensitivity analysis of the                               |
| 86  | calibration parameters. Section 5 shows how TRIX could be implemented for the whole                               |
| 87  | Adriatic Sea region and Section 6 presents the discussion and conclusions.                                        |
| 88  |                                                                                                                   |
| 89  | 2. TRIX equation and parameterizations                                                                            |
| 90  | The TRIX index was developed by Vollenweider et al. (1992) using data collected between                           |
| 91  | 1982 and 1993 by the "DAPHNE" oceanographic division of the Emilia-Romagna Regional                               |
| 92  | Environmental Protection Agency (hereafter referred as ARPAE-DAPHNE). Since 1971                                  |
| 93  | ARPAE-DAPHNE has been carrying out a monitoring program (Regione Emilia-Romagna,                                  |
| 94  | 1981-2013) covering the whole of the Emilia Romagna coastal region. The location of the                           |
| 95  | sampling stations is reported in Fig. 1.                                                                          |
| 96  | The TRIX index is based on four state variables $(n)$ , which are directly related to                             |
| 97  | productivity: chlorophyll-a (Chl, mg m <sup>-3</sup> ), oxygen as the absolute percentage deviation from          |
| 98  | oxygen saturation (DO, %), dissolved inorganic nitrogen (DIN, mg m <sup>-3</sup> ) and total                      |
| 99  | phosphorous ( <i>TP</i> , mg m <sup>-3</sup> ). In particular, $DIN = NO_3 + NO_2 + NH_4$ and $DO =  100 - Ox $ , |
| 100 | where $Ox$ is the oxygen saturation. Each state variable is scaled by the highest ( $U_i$ ) and the               |
| 101 | lowest $(L_i)$ values in the data time series and TRIX is defined as:                                             |
|     |                                                                                                                   |

Natural Hazards and Earth System Sciences Discussions

$$TRIX = \frac{k}{n} \sum_{i=1}^{n} \frac{(\log M_i - \log L_i)}{(\log U_i - \log L_i)}$$
 Eq. 1.1

where k=10 is another scaling factor, *n* is the number of state variables considered, and  $M_i$  are

the observed *Chl*, *DO*, *DIN* and *TP* values.

Vollenweider et al. (1998) further simplified the TRIX formula by assuming (on the basis of

the data used) that the difference ( $\log U - \log L$ ) was equal to 3 for all state variables.

Therefore, considering k=10, n=4 and the specific log  $L_i$  values (see Table III in Vollenweider

to et al., 1998), the TRIX formula was rewritten as follows:

$$TRIX = \frac{10}{12} \left[ \left( \log M_{chl} + 0.5 \right) + \left( \log M_{DO} + 1 \right) + \left( \log M_{DIN} - 0.5 \right) + \left( \log M_{TP} + 0.5 \right) \right]$$

or:

$$TRIX = \frac{\left[\log(Chl \times DO \times DIN \times TP) + 1.5\right]}{1.2}$$
Eq 1.2

Equation (1.2) gives the TRIX index currently used by ARPAE-Daphne and adopted by the 114 Italian national legislation (D.L. 260/2010). For the Italian coastal waters, TRIX values range from 0 to 10: 0 corresponds to extreme oligotrophic conditions; while 10 corresponds to 115 116 extreme eutrophic conditions. TRIX values have been further aggregated into four trophic 117 regimes (Rinaldi and Giovanardi, 2011): "Elevated", "Good", "Mediocre" and "Bad" (Table 118 1). Referring to Italian waters, TRIX values exceeding 6 are typical of highly productive 119 coastal areas, characterized by frequent episodes of anoxia at the sea bottom (Giovanardi and 120 Vollenweider, 2004). In the following sections we revise the TRIX index scaling parameters  $U_i$  and  $L_i$  on the basis 121 122 of different (observed and simulated) data sets, in order to evaluate the possibility of applying

the TRIX to numerical simulation data and extend its calculation to open ocean areas. The

assessment is carried out by closely comparing the simulated TRIX ("model TRIX") with

125 corresponding values from in situ observations ("in situ TRIX").

| 3. In situ and model data                                                                                 |
|-----------------------------------------------------------------------------------------------------------|
| 3.1 In situ data                                                                                          |
| The in situ data used in this paper were collected by the ARPAE-DAPHNE monitoring                         |
| program. We considered the 1982-1993 data time series originally used by Vollenweider et                  |
| al. (1998) to calibrate in situ TRIX and an additional recent time series covering the period             |
| 2001-2012 to validate the model TRIX. The monitoring grid considers 21 sampling stations                  |
| located along 8 transects perpendicular to the coast: 19 stations are coastal, extending from             |
| 500 m to 10 km offshore, while 2 stations are at 20 a km distance, sampling an open shelf                 |
| regime. All the stations are monitored weekly. ARPAE- DAPHNE divided the monitored                        |
| area into three sub-areas (A, B and C in Fig. 1) on the basis of the hydrological and trophic             |
| conditions (Montanari et al., 2006). Area A, is located immediately south of the Po delta and             |
| is directly affected by river runoff and nutrient load (see river Po in Fig.1); it is therefore           |
| characterized by enhanced primary production. Area B is a transition area, while area C is                |
| characterized by hydrographical conditions mainly governed by the large-scale basin                       |
| circulation.                                                                                              |
| The in situ TRIX was calculated for each station (using surface values of <i>Chl</i> , DO% <i>DIN</i> and |
| TP) and averaged over the transects and the three subareas. At the Porto Garibaldi and                    |
| Cesenatico transects (see Fig.1) TRIX was calculated with and without the open shelf                      |

stations.

*3.2 Numerical model data* 

The model data used in this study were produced by the three-dimensional coupled

- circulation-biogeochemical model consisting of the Princeton Ocean Model, POM (Blumberg
- and Mellor, 1987) and the Biogeochemical Flux Model-BFM (Vichi et al., 2007). The model

| 151 | was implemented in the Adriatic Sea at a horizontal resolution of about 2 km, and 27 sigma-      |  |  |  |
|-----|--------------------------------------------------------------------------------------------------|--|--|--|
| 152 | layers defined the vertical resolution (Clementi et al., 2010).                                  |  |  |  |
| 153 | BFM is a complex lower trophic marine biogeochemical model. It is a biomass based model,         |  |  |  |
| 154 | designed to simulate the main marine biogeochemical fluxes through a description of the          |  |  |  |
| 155 | ecological functions of the producers, decomposers and consumers and their specific trophic      |  |  |  |
| 156 | interactions in terms of basic elements (carbon, nitrogen, phosphorous, silicon and oxygen)      |  |  |  |
| 157 | flows. The biological constituents of the model are organized into Chemical Functional           |  |  |  |
| 158 | Families (CFFs) and Living Functional Groups (LFGs). CFFs are divided into organic (living       |  |  |  |
| 159 | and non-living) and inorganic compounds, which are measured in equivalents of major              |  |  |  |
| 160 | chemical elements or in molecular weight units. BFM receives information from the                |  |  |  |
| 161 | hydrodynamic model regarding temperature and salinity in order to calculate oxygen               |  |  |  |
| 162 | saturation.                                                                                      |  |  |  |
| 163 | The simulations were carried out for the period 1980-2010. Nutrients, oxygen and chlorophyll     |  |  |  |
| 164 | values were extracted at the model grid points nearest to the in situ sampling stations (grey    |  |  |  |
| 165 | shaded areas in Fig. 1). The model TRIX was then defined with $U_i$ and $L_i$ values computed    |  |  |  |
| 166 | using different time periods: 1982-1993 ("model TRIX"), 1991-2010 ("model TRIX 1"),              |  |  |  |
| 167 | 2001-2010 ("model TRIX 2") and 2006-2010 ("model TRIX 3"). The values obtained are               |  |  |  |
| 168 | reported in Table 2 and the model TRIX estimated from these different scaling parameters         |  |  |  |
| 169 | were compared with in situ values.                                                               |  |  |  |
| 170 |                                                                                                  |  |  |  |
| 171 | 4. Comparison of TRIX estimates from in-situ and simulated data.                                 |  |  |  |
| 172 | Figure 2 shows the comparison of the in situ and model TRIX considering the upper ( $U_i$ ) and  |  |  |  |
| 173 | lower ( $L_i$ ) values of each state variable obtained from the observed and simulated 1983-1992 |  |  |  |
| 174 | time series respectively.                                                                        |  |  |  |
| 175 | The figure clearly shows that the model TRIX lies in the same range of the in situ TRIX and      |  |  |  |
| 176 | the two time series show a distinct seasonal cycle with some degree of similarity. We            |  |  |  |

| 177 | computed the correlation coefficient using the simulated and the observed data series for the |
|-----|-----------------------------------------------------------------------------------------------|
| 178 | three sub-areas of the monitored region. The correlation coefficient values are reported in   |
| 179 | Table 3 and they show decreasing values from area A to C.                                     |
| 180 | Area A has the highest correlation values since it is the most eutrophic area, exhibiting the |
| 181 | highest TRIX values (> 6), depending on the direct influence of the fresh water runoff and    |
| 182 | nutrient load from the Po river delta (Fig. 1) immediately northward of area A. Areas B and   |
| 183 | C show progressively reduced TRIX values (Fig. 2B and C) denoting more oligotrophic           |
| 184 | conditions and reduced correlation values. This is mostly due to a model slightly             |
| 185 | underestimation of the index in area B between 2003 and 2006. Area C shows the lowest         |
| 186 | correlation because there is a temporal phase shift (of about two months) between the two     |
| 187 | time series between the 2005-2008. We presume that this is due to the particular Po river     |
| 188 | runoff and climatic conditions that are not well reproduced by the model.                     |
| 189 | In all the three study areas, TRIX increases between 2008-2010. Area A shows values above     |
| 190 | 6 ("bad" water quality conditions, see Table 1) in late winter-early spring and late summer-  |
| 191 | early autumn (Fig. 2). Areas B and C are characterized by TRIX values that indicate           |
| 192 | "Mediocre" (5 < TRIX < 6 in winter-spring), to "Good" (4 < TRIX < 5 in summer-autumn)         |
| 193 | conditions. However, some high in situ TRIX events (> 6) were recorded also in Area B         |
| 194 | during spring 2004 and 2010, while the "model TRIX" simulates values below 6 for all the      |
| 195 | period.                                                                                       |
| 196 | The TRIX index calculated as averages of each monitoring transect (Lido Volano,               |
| 197 | Casalborsetti, Ravenna, Lido Adriano, Rimini, Cattolica, Porto Garibaldi and Cesenatico, in   |
| 198 | Fig. 1) instead of averages over the areas A, B, C provides indications that are consistent   |
| 199 | with the previous results (see Fig 3). The correlations values relative to each transect are  |
| 200 | reported in Table 4 for all the transects and again highlight that the overall qualitative    |
| 201 | agreement between model and in situ TRIX, is reduced moving along an eutrophy to              |
| 202 | oligotrophy (north to south) gradient.                                                        |

| 203 | A preliminary conclusion is that model TRIX agrees, within the two standard deviations         |
|-----|------------------------------------------------------------------------------------------------|
| 204 | range, with the corresponding in situ index. In addition the model TRIX estimates appear       |
| 205 | more coherent with in situ estimations for trophic conditions characterized by a tendency to   |
| 206 | eutrophy.                                                                                      |
| 207 |                                                                                                |
| 208 | 4.2 Model TRIX sensitivity analysis                                                            |
| 209 | A sensitivity analysis was carried out by comparing the TRIX index calculated only with the    |
| 210 | coastal stations, up to 10 km distance from the coast (Fig. 4a and 4b), and the TRIX index     |
| 211 | computed also considering the two offshore stations, situated at 20 km off the coast for the   |
| 212 | Porto Garibaldi and Cesenatico transects (Fig. 4c and 4d). The correlation coefficient         |
| 213 | decreased for the Porto Garibaldi transect and did not change for the Cesenatico transect      |
| 214 | (Table 4). This further confirms that TRIX is most useful for coastal areas, under the direct  |
| 215 | influence of river runoff.                                                                     |
| 216 | Another sensitivity analysis was carried out by calculating the TRIX from simulated data       |
| 217 | using $U_i$ and $L_i$ scaling parameters from three more recent time series with a different   |
| 218 | temporal length years: 1991-2010 ("model TRIX 1"), 2001-2010 ("model TRIX 2") and              |
| 219 | 2001-2006 ("model TRIX 3") (see Table 2). The sensitivity of the TRIX to the three different   |
| 220 | sets of scaling values was estimated by computing the percentage difference (PD) between       |
| 221 | the model TRIX 1, 2 and 3 and the model TRIX obtained using the 1983-1992 data series:         |
| 222 | $PD_{i} = \frac{(TRIX_{i} - TRIX_{83-92})}{TRIX_{83-92}} $ 1.3                                 |
| 223 | where $i=1,2,3$ . The PD values are shown in Fig. 5 and illustrate the role on the TRIX values |
| 224 | of the time series extension and specific period chosen for the scaling parameters. A shorter  |
| 225 | time series (5 and 10 years) results in ~10% differences between estimates. Conversely the     |

- "model TRIX 1" computed with scaling parameters originating from a 20 years time series
- show values close to the model TRIX values calibrated with in situ data (Fig. 5).

10

- The correlation between the in situ TRIX and the TRIX calculated from model simulations
- ("model TRIX" and "model TRIX 1, 2 and 3") varied depending on the time period
- considered (Table 3). Generally, area A was the best-fitted area, with high correlation values
- for all the time periods studied. Correlation values decreased in areas B and C when we
- considered short time period data (5-10 years). However a high correlation was observed, in
- all the three areas, between the "in situ TRIX" and the "model TRIX 1" calculate using  $U_i$
- and  $L_i$  values from a long time series (20 years) (Table 3).
- In conclusion this sensitivity study shows that TRIX values become independent on the
- scaling parameter values if the time series is longer than 10 years. In all cases, model values
- are better correlated to in situ ones in eutrophic as compared to quasi-oligotrophic areas and
- in near shore than offshore areas.

## 240 5. Adriatic Sea model TRIX

- Finally, the model TRIX was computed for the whole Adriatic and Northern Ionian Sea. The
  monthly means computed for the time period 2001-2010 are shown in Fig. 6.
- There is a sharp contrast between the coastal northwestern shores, the southeastern shelves
- and the open sea areas. The northwestern Adriatic Sea shows the highest TRIX values (> 6),
- corresponding to eutrophic conditions that could be related to possible anoxia/hypoxia
- events. The TRIX values progressively decrease along the western Adriatic coast, with
- values ranging between 5 and 6 on the Emilia-Romagna coast, and values 

11

In this paper we have shown a methodology to generalize the TRIX index to an entire sea basin region and to simulation model data. First the TRIX index equation was re-written with 255 general scaling parameters as in Vollenweider et al. (1992). Each state variable is scaled by 256 257 the highest  $(U_i)$  and the lowest  $(L_i)$  values in the data time series and in our paper we tested the importance of the length and the specific period used to evaluate these scaling 258 259 parameters. 260 The analysis was based on a comparison of in situ data derived TRIX and the model TRIX in 261 the Emilia- Romagna coastal strip. First of all, the results indicated the generally significant 262 potential skill of the model in replicating the observed index, provided that the TRIX scaling parameters are computed from the simulated data set itself. The comparison also indicated 263 264 that in a statistical sense, the model's replication skill decreases consistently with the trophic characteristics tendency towards oligotrophic conditions. This is indicated by the decreasing 265 266 correlation values from area A to areas B and C. This obviously merits further investigation 267 as it is probably related to the main driver controlling the local environmental conditions 268 (external input vs. circulation and vertical structure dynamics). 269 The sensitivity analysis to the extension and specific time series used to evaluate the scaling 270 parameters indicated that for time series longer than 10 years the results were insensitive to 271 the  $(U_i)$  and  $(L_i)$  values. This is because a long data series is more likely to encompass a 272 wide range of events and better estimate extremes. 273 When implemented in the whole Adriatic Sea basin scale, the model TRIX produced the 274 sharp transition from the eutrophic oriented conditions of the coastal domain, to the 275 oligotrophic conditions that characterise the pelagic domain. 276 Numerical simulations can therefore represent an important support for monitoring activities, i.e. they will allow to extend the use of TRIX to much larger areas where in situ sampling 277 278 activities are difficult to implement.