# Peer review of "(untitled)"

_Natural Hazards and Earth System Sciences, 2016_

## Referee Comment (RC1) · Anonymous Referee #1 · 18 Apr 2016

Rome, 14 Apr 2016

In the following I report the requested comments, as I was required as an appointed referee, in order to support the review process of the manuscript:

**A Generalization of the TRIX trophic index to the Adriatic Sea basin.**

Author(s): E. Fiori et al. - MS No.: nhess-2016-69 - MS Type: Research article
Special Issue: Situational sea awareness technologies for maritime safety and marine environment protection

A.      With reference to the criteria recommended by the Editor, and expressing an overall judgment, this work is surely to be considered as an excellent work, both for its Scientific Significance and for the related Scientific Quality. It represents undoubtedly a serious contribution to the understanding "of natural and man-made hazards" in coastal-marine areas, to say the study of coastal-marine Eutrophication, by proposing new ideas and useful methodological approaches. In this respect, the interest for the work presented here, derives mainly from the mathematical approach adopted, which allowed to process and compare long historical data series available for the Adriatic, to the output of simulation models. The results are discussed in an appropriate and balanced way (i.e. clarity of concepts and discussion, including appropriate literature references).

As for the **Presentation Quality**, in this case also the judgement is to be considered  "good": results and conclusions are presented in a clear, concise, and well-structured way (number and quality of figures/tables, appropriate use of technical and English language, simplicity and good understanding of the language used).

B.      Now I enter into the merits of the work, not only  to contribute to the discussion, but also to point out the need for some clarification in the text.

1.          *"The final results of this paper present a new TRIX index formulation that could be used to assess the GES of coastal and marine waters in terms of eutrophication status."*  (rows 80-82 of the manuscript).

The reference to the Marine Strategy Framework Directive is all too clear. However, it should be noted that both the EU Directives, the MSFD and also the WFD 2000/60 EU, in general give priority to the Biological Elements and they assume an approach based on DPSIR model (Driving forces, Pressures, Status, Impacts, Responses). Under this aspect the TRIX has been defined as a linear combination of the four fundamental trophic status indicators (chlorophyll, nitrogen, phosphorus and oxygen deviation from saturation). As such, TRIX represents a mixing of pressure (N and P) and impact (Chl-a and oxygen deviation) indicators and finally it does not result fully consistent with the MSFD "philosophy", nor with the WFD requirements.

Nevertheless, it can work as an excellent control variable  and, in this light, it has already been considered by the Italian legislation that received  the WFD  directive, just as a variable supporting  the judgment of ecological status. (Cfr. Decree 260/2010 "Classification criteria" at  Cap. A.4.3.2: Technical criteria for classification on the basis of physico-chemical and hydromorphological quality elements in support. In particular Table 4.3.2./c provides class boundaries, expressed in terms of TRIX units, that cannot be exceeded if the resulting ecological status, based on the related Biological Quality Elements, belongs to the "good"class.)

2.          *"However, the general methodology for constructing TRIX for the different areas has not been clarified. In order to apply TRIX to different areas a precise evaluation of the scaling parameters is required."*  (rows 73-75 of the manuscript).

In the cited work of Giovanardi and Vollenweider (2004), on the meaning of the TRIX it is reported:

"…*a purely numeric scale that scores the trophic properties of surface and subsurface seawaters, station by station and/or sequential in time, would be preferable to some preconceived categorical denominators*."

Clearly this definition is limitative in the sense that it provides only one facet, while the broad array of tropho-dynamic processes and the related biological diversities that characterize any marine-coastal area, will have to be added by appropriate research.

On the other hand, **the approach is general, not geographically bound,** but on which the mentioned arrays can be built as a platform.
On this basis it was defined the trophic scale of the TRIX values, also reported in the present manuscript as Table 1 (pag. 23). The aim was to define a general reference system in order to make comparisons between different sea areas and / or between different seasonal periods on the same sea zone. It is clear that these comparisons will be possible,

only if the measuring tool used (i.e. the TRIX Index), was built with a unique criterion, namely, always using the same scales to define the range of variation of each component.

On the other hand (rows 216-227: Sensitivity analysis), the authors make abundantly clear why they have chosen different scales for TRIX components, depending on different areas and/or different temporal series The pursued aim is certainly to be appreciated, being finalized not only to test the index used in different areas (affected and not affected directly by river inputs), but also to demonstrate the goodness of the general approach followed in the formulation of TRIX Index.

3. *"Numerical simulations can therefore represent an important support for monitoring activities, i.e. they will allow to extend the use of TRIX to much larger areas where in situ sampling activities are difficult to implement. "* (Rows 276-278 - Conclusion)

I think this is really the main scientific merit of all this work, that comes not only from the analysis of long time series of "real" data, but also by the possibility of generating simulated time series as output from numerical models.
In this regard I wish to emphasize another merit of the work, it presents the results of the simulation models, without going too over the more technical and mathematical aspects and referring the interested reader to appropriate literature.

This has certainly helped to ease the reading of the work, allowing even those who are not expert in mathematical modeling to understand the results and the objectives pursued by the numerical simulations.

One last observation. The authors used the results of the models (i.e. the values of the components of the TRIX provided by simulation models), to describe the behavior of the index TRIX as such, in its temporal and spatial variability. But they omitted to consider the potential changes of the percentage contributions of the components of trix, in time and space.

This datum could be very interesting. The weight of each component in the index formulation is in fact to be related to the so-called "efficiency" of the marine and coastal systems to use nutrients to produce phytoplankton biomass.
Of course, to concur in the formulation of the TRIX may prevail the components relating to the active phytoplankton biomass (Chl and Oxygen deviation from 100%), or nutritional factors, (N and P). In the latter case, the system does not use all the available nutrients and thus shows little "efficiency."

As discussed in the paper mentioned above (Giovanardi and Vollenweider, 2004), coastal systems of the Tyrrhenian and Adriatic Seas they differ primarily because of their peculiar "efficiency" in utilizing nutrients. But perhaps these aspects deserve to be treated and deepened, following the same approach already followed here, in a later work.

4. *The TRIX index is based on four state variables (n), which are directly related to productivity: chlorophyll-a (Chl, mg m$^{-3}$), oxygen as the absolute percentage deviation from oxygen saturation (DO, %), dissolved inorganic nitrogen (DIN, mg m$^{-3}$) and total phosphorous (TP, mg m$^{-3}$). In particular, DIN = NO$_3$ + NO$_2$ + NH$_4$ and DO = abs( 100-Ox), where Ox is the oxygen saturation.* (Rows 96 -100).

Some imprecision may lead to confusion, as it has already happened on other occasions.
Is it correct to indicate the units of measurement expressed as weights (mg m$^{-3}$) for Chlorophyll and Nutrients. But be aware that the relation DIN = NO$_3$ + NO$_2$ + NH$_4$, as it is reported in the text, is stoichiometrically incorrect. It would be correct if the units were all expressed in mM m$^{-3}$. The correct form is therefore the following:
DIN = N-NO$_3$ + N-NO$_2$ + N-NH$_4$,
that is to say, at risk of becoming pedantic, DIN, Dissolved Inorganic Nitrogen in weight, corresponds to the sum of Nitric Nitrogen, Nitrous Nitrogen and Ammonia Nitrogen.

Recalling the very positive overall assessment of the work, as already expressed at the beginning, I believe that the authors should take into account these observations and adjust the text of the manuscript, where necessary.

---

## Referee Comment (RC2) · Anonymous Referee #2 · 19 May 2016

*Review of "A Generalization of the TRIX trophic index to the Adriatic Sea basin"* (by Dr. M. Fiori and co-authors (Manuscript under review for journal Nat. Hazards Earth Syst. Sci. doi:10.5194/nhess-2016-69, 2016)

The paper "A Generalization of the TRIX trophic index to the Adriatic Sea basin" compares TRIX index values from in situ data with those from a coupled physics and biogeochemical numerical model implemented in the overall Adriatic Sea. The study aims to demonstrate the compatibility of the model with the in situ TRIX and the necessity to have time series longer than 10 years to evaluate properly the scaling parameters.

The first scope of this paper is very interesting and I think that estimating the trophic index from model outputs could be very useful in supporting policy. However, it should be stated here that the authors are using only one model with a single realization. This is fine with the scope of this study, but it is a limitation for the evaluation of model uncertainty.

Regarding the second object of this study, the demonstration that a sample longer than 10 years is needed for a good estimation of the TRIX scaling parameters is an obvious concept in statistics and in particular in Extreme Values Theory (EVT, see Coles 2001).

By definition the TRIX scaling parameters are yearly (or seasonal) maxima and minima in a period of a fixed length (e.g. 10, 20, 30... years). These parameters follow a Genaralized Extreme Values (GEV) distribution (See Coles et al 2001). A robust estimation of the GEV parameters is based on large-sample theory (asymptotic property), meaning that the times series of the extremes has to be as longer as possible to have stable estimates across periods with different lengths.

As a conclusion, this manuscript can be published after major revision. It is of great interest regarding the comparison between model outputs and observations, however, it has to be revised in all the sections aiming to demonstrate the obvious conclusion that a longer time series provide more robust index estimates. Please, see further suggestions below.

**Specific comments**

- l.1 I think that the title should be revised accordingly to comments provided above. A possible title could be: "Observed and simulated TRIX trophic index values to the Adriatic Sea basin"

- l.25 The main scope of the MWFD is to achieve Good Environmental Status (GES) of the EU's marine waters by 2020" Please, rephrase this

sentence by reporting the main scope of the Marine strategy and the importance of environmental composite indices in assessing the Good Environmental Status (GES).

- l.29-31 it seems that the authors have modified the previous TRIX definition by defining the TRIX parameters in a different way. As already explained above this is not the case of this study. The authors have only calculated the TRIX parameters with a record length longer than 10 years. Please, remove this sentence or rephrase it accordingly.

- l.36 I would suggest to replace "demonstrates" with "shows"

- l.37 please make this sentence more general. I would suggest something like this: "and as the length of the time series is relevant to get robust index estimates.

- l.54 replace "hence indicators" with "hence composite indicators"

- l.73 this sentence is unclear to me. As the authors reported above the TRIX index has been defined in Volleweider et al. 1998.

- l.74-l77 Extending the TRIX calculation from a period of 10 to one of 20 years does not mean to review the methodology. Please review accordingly.

- l.78-l87 As explained above I do not think this paper present a new TRIX index. Accordingly with Vollenweider et al 1998 the parameter L and M of the TRIX index are defined in log units. The limits for each log transformed variable composing the TRIX index are fixed a priori (see Vollenweider et al Table 3). These limits are fixed a priori from a standard normal distribution of log-transformed variables, being independent from the length of the time series. Of course the correspondence between the fixed limits in log units reported in Vollenweider et al 1998 (see Table 3) and the physical value of the variable will change with the length of the data record. In Vollenweider et al the authors have used a time series of 12 years (1982-1993) because a dataset with longer record length was not available. However, they never wrote in their manuscript that the TRIX limits parameters are fixed with respect a time series of 10 or 12 years. I strongly suggest to review this paper accordingly to this point and to specify that the aims is not to define/modify the TRIX, but to apply the TRIX to observed and model data. This could be very useful for policy makers in order to adopt new strategies.

- l.121-l.125 please review this para accordingly to my comments above (e.g. see l.78-l.87)

- l.180-l.188 Please, specify which correlation coefficient has been used here. I think it is Pearson because log transformed data follow a normal distribution, however it would be better to report this information. Moreover a level of significance (p-value) for correlation should be reported.

- l.235-l.238 Please review this sentence accordingly to comments above (e.g. see l.78-l.87). We need a very large sample size in order to get robust estimates. Recently Sippel et al GRL have shown as $\mu$ and $\sigma$ estimates could be biased by using a time series of 30-year. However this is not the case of the TRIX parameters, as explained above they are fixed a priori on log transformed and standardized variables. To simplify calculation, the ranges have been standardized to 3 log units (see Vollenweider et al 1998).

- Discussion and Conclusions: please review this section by following my comments above. I would focus on the most interesting part of this manuscript which is comparison between simulated and observed TRIX. Moreover, please add somewhere in the text a sentence specifying that in this study only one model with a single realization has been used, limiting the possibility to estimate model uncertainty.

- Suggestions: what about to use a reference period of 30-year to define the TRIX parameters as done for climate extreme indices ($http : //etccdi.pacificclimate.org/list_27_indices.shtml$)?
  A reference period of at least 30-year is suggested by the World Meteorological Organization in order to have robustness of the indices with respect to the length of the time series.
  As an example following Russo et al 2015 using a non-parametric approach to standardize hot days in a heatwave, the authors could define lower and upper TRIX limits on 30-year time series as the 25th and 75th percentile, respectively.

**Refrences:**

Coles, S. (2001), An Introduction to Statistical Modeling of Extreme Values, 208 pp., *Springer, Berlin*.

Sippel, S., J. Zscheischler, M. Heimann, F. E. L. Otto, J. Peters, and M. D. Mahecha (2015), Quantifying changes in climate variability and extremes: Pitfalls and their overcoming, Geophys. Res. Lett., 42, 99909998.

Russo S, Sillmann J and Fischer E M 2015 Top ten European heat-waves since 1950 and their occurrence in the coming decades Environ. Res. Lett. 10 124003

---

## Author Comment (AC1) · 5 Jul 2016

Below we report our answers to all reviewer's comments and suggestions.

Rome, 14 Apr 2016

In the following I report the requested comments, as I was required as an appointed referee, in order to support the review process of the manuscript: A Generalization of the TRIX trophic index to the Adriatic Sea basin. Author(s): E. Fiori et al. - MS No.: nhess-2016-69 - MS Type: Research article

Special Issue: Situational sea awareness technologies for maritime safety and marine environment protection

A. With reference to the criteria recommended by the Editor, and expressing an overall judgment, this work is surely to be considered as an excellent work, both for its Scientific Significance and for the related Scientific Quality. It represents undoubtedly a serious contribution to the understanding "of natural and man-made hazards" in coastal-marine areas, to say the study of coastal-marine Eutrophication, by proposing new ideas and useful methodological approaches. In this respect, the interest for the work presented here, derives mainly from the mathematical approach adopted, which allowed to process and compare long historical data series available for the Adriatic, to the output of simulation models. The results are discussed in an appropriate and balanced way (i.e. clarity of concepts and discussion, including appropriate literature references).

We thank the reviewer for the nice comment.

As for the Presentation Quality, in this case also the judgement is to be considered "good": results and conclusions are presented in a clear, concise, and well-structured way (number and quality of figures/tables, appropriate use of technical and English language, simplicity and good understanding of the language used).

Again thank you very much.

B. Now I enter into the merits of the work, not only to contribute to the discussion, but also to point out the need for some clarification in the text.

1. "The final results of this paper present a new TRIX index formulation that could be used to assess the GES of coastal and marine waters in terms of eutrophication status." (rows 80-82 of the manuscript).

The reference to the Marine Strategy Framework Directive is all too clear. However, it should be noted that both the EU Directives, the MSFD and also the WFD 2000/60 EU, in general give priority to the Biological Elements and they assume an approach based on DPSIR model (Driving forces, Pressures, Status, Impacts, Responses). Under this aspect the TRIX has been defined as a linear combination of the four fundamental trophic status indicators (chlorophyll, nitrogen, phosphorus and oxygen deviation from saturation). As such, TRIX represents a mixing of pressure (N and P) and impact (Chl-a and oxygen deviation) indicators and finally it does not result fully consistent with the MSFD "philosophy", nor with the WFD requirements.

Nevertheless, it can work as an excellent control variable and, in this light, it has already been considered by the Italian legislation that received the WFD directive, just as a variable supporting the judgment of ecological status. (Cfr. Decree 260/2010 "Classification criteria" at Cap. A.4.3.2: Technical criteria for classification on the basis of physico- chemical and hydromorphological quality elements in support. In particular Table 4.3.2./c provides class boundaries, expressed in terms of TRIX units, that cannot be exceeded if the resulting ecological status, based on the related Biological Quality Elements, belongs to the "good"class.)

The reviewer is entirely correct in stating that the entire MSFD approach is based on the DPSIR conceptual model, while the TRIX index takes into consideration and integrate elements that can be regarded as pressure and state. Therefore, TRIX is not fully consistent with the MSFD approach. However, considering the EU GES criteria and descriptors (in particular the Eutrophication Descriptors), as defined in the pertinent EU official documents, it appears as quite straightforward that an index such as the TRIX is well suited to provide an assessment of the Environmental status with respect to "Eutrophication", and this is exactly because it provides a synthesis between elements of the DPSIR conceptual model:

Effectively we agree that our wording was probably a little bit confusing. Therefore we are reading to modify the original manuscript (lines 50-53 of the submitted manuscript) text with the following sentence:

*The EU-Marine Strategy Framework Directive (MSFD, 2008/56/EC) address the overall state of the marine environment with a DPSIR (Driver, Pressure, Impact State Response) conceptual approach and explicitly considers eutrophication as a crucial process that can alterate the "Good Environmental Status" (GES) of European coastal waters. A synthetic indicator of the environmental state of the coastal ocean with respect to the eutrophication process integrating elements of the DPSIR methodology is therefore neededl to provide an objective assessment of the environmental*

*state. Furthermore, it provides elements for the implementation of an ecosystem-based strategy for the achievement and maintenance of GES.*

We will correct accordingly line 80-82 of the submitted manuscript:

*The final results of this paper could be used as criterion to classify the marine ecosystem (D.L 260/2010), providing class boundaries expressed as TRIX units (Table 4.3.2./c). Furthermore, the model ecosystem can represent an important support for monitoring activities, i.e. to extend the use of TRIX to larger areas where in situ sampling activities are difficult to implement.*

2. "However, the general methodology for constructing TRIX for the different areas has not been clarified. In order to apply TRIX to different areas a precise evaluation of the scaling parameters is required." (rows 73-75 of the manuscript).

In the cited work of Giovanardi and Vollenweider (2004), on the meaning of the TRIX it is reported: "...a purely numeric scale that scores the trophic properties of surface and subsurface seawaters, station by station

and/or sequential in time, would be preferable to some preconceived categorical denominators."

Clearly this definition is limitative in the sense that it provides only one facet, while the broad array of tropho-dynamic processes and the related biological diversities that characterize any marine-coastal area, will have to be added by appropriate research.

On the other hand, the approach is general, not geographically bound, but on which the mentioned arrays can be built as a platform. On this basis it was defined the trophic scale of the TRIX values, also reported in the present manuscript as Table 1 (pag. 23). The aim was to define a general reference system in order to make comparisons between different sea areas and / or between different seasonal periods on the same sea zone. It is clear that these comparisons will be possible, only if the measuring tool used (i.e. the TRIX Index), was built with a unique criterion, namely, always using the same scales to define the range of variation of each component.

We agree with the reviewer that our wording was not precise and it could generate misunderstanding. It is clear to us that the definition of the TRIX index is not geographically bound and the aim of the original TRIX developers was the definition of a general procedure to assess the trophic state of the coastal waters. Our original meaning was to underline how the TRIX computation from the output of a numerical model should necessarily consider the unavoidable differences between the "model" and the observed system.

This called for a reconsideration of the TRIX original formula (i.e. eliminating the simplifying assumptions made by the specific TRIX development for the Emilia-Romagna area) and for a rescaling of the TRIX parameters so that the upper and lower limits of the biogeochemical properties entering the TRIX equation were made consistent with the model dynamics.

On the basis of this we plan to rewrite the pertinent section of the manuscript as follow (modification of lines 73-87 of the original submission):

*In this paper we compared TRIX in situ with model simulations for long data series in different coastal and open ocean areas. The specific objectives of our work are: (1) to adapt the TRIX generic equation (1.1) to numerical ecosystem model simulation data, (2) to validate the "model TRIX" with in situ data in different areas and time series, (3) to apply the TRIX generic equation to other coastal and open ocean areas in the entire Adriatic Sea.*

*The final results of this paper could be used as criterion to classify the marine ecosystem (D.L 260/2010), providing class boundaries expressed as TRIX units (Table 4.3.2./c). Furthermore, the model ecosystem can represent an important support for monitoring activities, i.e. to extend the use of TRIX to larger areas where in situ sampling activities are difficult to implement.*
*Section 2 describes the TRIX equation and its calibration parameters for the model simulations. Section 3 illustrates the in situ and simulation data used for the evaluation of TRIX and its calibration. Section 4 compares the "in situ TRIX" and "model TRIX", and the sensitivity analysis of the calibration parameters. Section 5 shows how TRIX could be implemented for the whole Adriatic Sea region and Section 6 presents the discussion and conclusions.*

On the other hand (rows 216-227: Sensitivity analysis), the authors make abundantly clear why they have chosen different scales for TRIX components, depending on different areas and/or different temporal series The pursued aim is certainly to be appreciated, being finalized not only to test the index used in different areas (affected and not affected directly by river inputs), but also to demonstrate the goodness of the general approach followed in the formulation of TRIX Index.

The sensitivity analysis allowed us to demonstrate that, despite the unavoidable model deficiencies, model predictions have a practical value: once carried out the required re-scaling, the model TRIX index partially reproduces the observed TRIX. This implies that the model has a potential skill for reproducing an integrated environmental index. Furthermore, it shows the importance to have long time data series in order to reduce the possible simulation or systematic error bias of the simulations.

3. "Numerical simulations can therefore represent an important support for monitoring activities, i.e. they will allow to extend the use of TRIX to much larger areas where in situ sampling activities are difficult to implement. " (Rows 276-278 - Conclusion)

I think this is really the main scientific merit of all this work, that comes not only from the analysis of long time series of "real" data, but also by the possibility of generating simulated time series as output from numerical models. In this regard I wish to emphasize another merit of the work, it presents the results of the simulation models, without going too over the more technical and mathematical aspects and referring the interested reader to appropriate literature.

This has certainly helped to ease the reading of the work, allowing even those who are not expert in mathematical modeling to understand the results and the objectives pursued by the numerical simulations.

Thank you very much for this nice comment.

One last observation. The authors used the results of the models (i.e. the values of the components of the TRIX provided by simulation models), to describe the behavior of the index TRIX as such, in its temporal and spatial variability. But they omitted to consider the potential changes of the percentage contributions of the components of trix, in time and space.

We agree with the reviewer and we are going to add a small section describing two new figures showing the individual percentage contributions of each variable composing the TRIX to the complete final value of the index. Individual percentage contribution has been computed for the "observed" and "simulated" TRIX and a full comparison is provided and discusses.

This datum could be very interesting. The weight of each component in the index formulation is in fact to be related to the so-called "efficiency" of the marine and coastal systems to use nutrients to produce phytoplankton biomass. Of course, to concur in the formulation of the TRIX may prevail the components relating to the active phytoplankton biomass (Chl and Oxygen deviation from 100%), or nutritional factors, (N and P). In the latter case, the system does not use all the available nutrients and thus shows little "efficiency."

As discussed in the paper mentioned above (Giovanardi and Vollenweider, 2004), coastal systems of the Tyrrhenian and Adriatic Seas they differ primarily because of their peculiar "efficiency" in utilizing nutrients. But

perhaps these aspects deserve to be treated and deepened, following the same approach already followed here, in a later work.

We agree that the "efficiency" pattern is very important to study more in details the marine and coastal ecosystem. Then, we are going to calculate the efficiency coefficient from the numerical and the observational data sets according to the definition proposed by Giovanardi and Vollenweider, (2004).

4. The TRIX index is based on four state variables (n), which are directly related to productivity: chlorophyll-a (Chl, mg m$^{-3}$), oxygen as the absolute percentage deviation from oxygen saturation (DO, %), dissolved inorganic nitrogen (DIN, mg m$^{-3}$) and total phosphorous (TP, mg m$^{-3}$). In particular, DIN = NO3 + NO2 + NH4 and DO = abs( 100-Ox), where Ox is the oxygen saturation. (Rows 96 -100).

Some imprecision may lead to confusion, as it has already happened on other occasions. Is it correct to indicate the units of measurement expressed as weights (mg m$^{-3}$) for Chlorophyll and Nutrients. But be aware that the relation DIN = $NO_3$ + $NO_2$ + $NH_4$, as it is reported in the text, is stoichiometrically incorrect. It would be correct if the units were all expressed in mM m$^{-3}$. The correct form is therefore the following: DIN = N-$NO_3$ + N-$NO_2$ + N-$NH_4$, that is to say, at risk of becoming pedantic, DIN, Dissolved Inorganic Nitrogen in weight, corresponds to the sum of Nitric Nitrogen, Nitrous Nitrogen and Ammonia Nitrogen.

Thanks for noticing this and we apologize for the confusion. We will correct the text accordingly.

Recalling the very positive overall assessment of the work, as already expressed at the beginning, I believe that the authors should take into account these observations and adjust the text of the manuscript, where necessary.

---

## Author Comment (AC2) · 5 Jul 2016

Below we report our answers to all reviewer's comments and suggestions.

Review of "A Generalization of the TRIX trophic index to the Adriatic Sea basin" (by Dr. M. Fiori and co-authors (Manuscript under review for journal Nat. Hazards Earth Syst. Sci. doi:10.5194/nhess-2016-69, 2016)

The paper "A Generalization of the TRIX trophic index to the Adriatic Sea basin" compares TRIX index values from in situ data with those from a coupled physics and biogeochemical numerical model implemented in the overall Adriatic Sea. The study aims to demonstrate the compatibility of the model with the in situ TRIX and the necessity to have time series longer than 10 years to evaluate properly the scaling parameters.

The first scope of this paper is very interesting and I think that estimating the trophic index from model outputs could be very useful in supporting policy. However, it should be stated here that the authors are using only one model with a single realization. This is fine with the scope of this study, but it is a limitation for the evaluation of model uncertainty.

We agree with the reviewer that in the future we could also use a super-ensemble model estimation in order to better map the model uncertainties. We will mention this in the conclusions.

Regarding the second object of this study, the demonstration that a sample longer than 10 years is needed for a good estimation of the TRIX scaling parameters is an obvious concept in statistics and in particular in Extreme Values Theory (EVT, see Coles 2001).

We are aware of the obvious statistical meaning of the statement. However, we mentioned this in an effort to establish the minimal length of a simulation needed in order to provide adequate scaling parameter.

By definition the TRIX scaling parameters are yearly (or seasonal) maxima and minima in a period of a fixed length (e.g. 10, 20, 30... years). These parameters follow a Genaralized Extreme Values (GEV) distribution (See Coles et al 2001). A robust estimation of the GEV parameters is based on large-sample theory (asymptotic property), meaning that the times series of the extremes has to be as longer as possible to have stable estimates across periods with different lengths.

As a conclusion, this manuscript can be published after major revision. It is of great interest regarding the comparison between model outputs and observations, however, it has to be revised in all the sections aiming to demonstrate the obvious conclusion that a longer time series provide more robust index estimates. Please, see further suggestions below.

Specific comments

• l.1 I think that the title should be revised accordingly to comments provided above. A possible title could be: "Observed and simulated TRIX trophic index values to the Adriatic Sea basin"

  We agree and we modify the title as suggested.

• l.25 The main scope of the MWFD is to achieve Good Environmental Status (GES) of the EU's marine waters by 2020" Please, rephrase this sentence by reporting the main scope of the Marine strategy and the importance of environmental composite indices in assessing the Good Environmental Status (GES).

We agree and we will change this phrase according to the MSFD legislation. According also to the other reviewer we will modified also line lines 50-53 of the submitted manuscript) text with the following sentence:

*The EU-Marine Strategy Framework Directive (MSFD, 2008/56/EC) address the overall state of the marine environment with a DPSIR (Driver, Pressure, Impact State Response) conceptual approach and explicitly considers eutrophication as a crucial process that can alterate the "Good Environmental Status" (GES) of European coastal waters. A synthetic indicator of the environmental state of the coastal ocean with respect to the eutrophication process integrating elements of the DPSIR methodology (as explained below) is therefore very useful to provide an objective assessment of the environmental state. Furhtermore, it provides elements for the implementation of an ecosystem-based strategy for the achievement and maintenance of GES.*

• l.29-31 it seems that the authors have modified the previous TRIX definition by defining the TRIX parameters in a different way. As already explained above this is not the case of this study. The authors have only calculated the TRIX parameters with a record length longer than 10 years. Please, remove this sentence or rephrase it accordingly.

Yes the sentence was confusing. We remove it from the revised manuscript version.

- l.36 I would suggest to replace "demonstrates" with "shows"

    Modified accordingly.

- l.37 please make this sentence more general. I would suggest something like this: "and as the length of the time series is relevant to get robust index estimates.

    Modified accordingly.

- l.54 replace "hence indicators" with "hence composite indicators"

    Modified accordingly.

- l.73 this sentence is unclear to me. As the authors reported above the TRIX index has been defined in Vollenweider et al. 1998.

The reviewer is right; the sentence was confusing and has been substantially modified. On the basis of this comment, and according to reviewer N1, we plan to rewrite the pertinent section of the manuscript as follow (modification of lines 73-87 of the original submission):

*In this paper we compared TRIX in situ with model simulations for long data series in different coastal and open ocean areas. The specific objectives of our work are: (1) to adapt the TRIX generic equation (equation 1.1) to numerical ecosystem model simulation data (2) to validate the "model TRIX" with in situ data in different areas and time series (3) to apply the TRIX generic equation to other coastal and open ocean areas in the entire Adriatic Sea*

*The final results of this paper could be used as criterion to classify the marine ecosystem (D.L 260/2010), providing class boundaries expressed as TRIX units (Table 4.3.2./c). Furthermore, the model ecosystem can represent an important support for monitoring activities, i.e. to extend the use of TRIX to larger areas where in situ sampling activities are difficult to implement.*

*Section 2 describes the TRIX equation and its calibration parameters for the model simulations. Section 3 illustrates the in situ and simulation model data used for the evaluation of TRIX and its calibration. Section 4 compares the "in*

*situ TRIX" and "model TRIX", and the sensitivity analysis of the calibration parameters. Section 5 shows how TRIX could be implemented for the whole Adriatic Sea region and Section 6 presents the discussion and conclusions.*

l.74-l77 Extending the TRIX calculation from a period of 10 to one of 20 years does not mean to review the methodology. Please review accordingly.

We have revised the sentence.

l.78-l87 As explained above I do not think this paper present a new TRIX index. Accordingly with Vollenweider et al 1998 the parameter L and M of the TRIX index are defined in log units. The limits for each log transformed variable composing the TRIX index are fixed a priori (see Vollenweider et al Table 3). These limits are fixed a pri- ori from a standard normal distribution of log-transformed variables, being independent from the length of the time series. Of course the correspondence between the fixed limits in log units reported in Vollenweider et al 1998 (see Table 3) and the physical value of the variable will change with the length of the data record. In Vollenweider et al the authors have used a time series of 12 years (1982-1993) because a dataset with longer record length was not available. However, they never wrote in their manuscript that the TRIX limits parameters are fixed with respect a time series of 10 or 12 years. I strongly suggest to review this paper accordingly to this point and to specify that the aims is not to define/modify the TRIX, but to apply the TRIX to observed and model data. This could be very useful for policy makers in order to adopt new strategies.

Yes, our wording was effectively misleading. Then, according also to reviewer N1, we have redefined the text by stressing that the aim of the work is not to redefine TRIX, but evaluate how TRIX could be sensibly reproduced using data from a numerical model.

- l.121-l.125 please review this para accordingly to my comments above (e.g. see l.78-l.87)

We will modify this part accordingly to the comments.

- l.180-l.188 Please, specify which correlation coefficient has been used here. I

think it is Pearson because log transformed data follow a normal distribution, however it would be better to report this information. Moreover a level of significance (p-value) for correlation should be reported.

We used Pearson correlation coefficient. We will specify the information in the text and we will add the p-value to table 3 and 4.

- l.235-l.238 Please review this sentence accordingly to comments above (e.g. see l.78-l.87). We need a very large sample size in order to get robust estimates. Recently Sippel et al GRL have shown as $\mu$ and $\sigma$ estimates could be biased by using a time series of 30-year. However this is not the case of the TRIX parameters, as explained above they are fixed a priori on log transformed and standardized variables. To simplify calculation, the ranges have been standardized to 3 log units (see Vollenweider et al 1998).

  Yes, we will modified this sentence as reported below:

  *"The results of the sensitivity analysis, focused on the assessment of the model capability to provide an useful TRIX index, suggest the use of simulated time series of at least ten years to obtain the upper and lower limits for the state variables involved in the TRIX computation."*

- Discussion and Conclusions: please review this section by following my comments above. I would focus on the most interesting part of this manuscript which is comparison between simulated and observed TRIX. Moreover, please add somewhere in the text a sentence speci- fying that in this study only one model with a single realization has been used, limiting the possibility to estimate model uncertainty.

  We will correct these sections according to the comments.

- Suggestions: what about to use a reference period of 30-year to de- fine the TRIX parameters as done for climate extreme indices (http : //etccdi.pacificclimate.org/list$_2$7¡ndices.shtml)? A reference period of at least 30-year is suggested by the World Mete- orological Organization in order to have robustness of the indices with respect to the length of the time series. As an example following Russo et al 2015 using a non-parametric ap- proach to standardize hot days in a heatwave, the authors

could define lower and upper TRIX limits on 30-year time series as the 25th and 75th percentile, respectively.

- Refrences:  Coles, S. (2001), An Introduction to Statistical Modeling of Extreme Values, 208 pp., Springer, Berlin.

Sippel, S., J. Zscheischler, M. Heimann, F. E. L. Otto, J. Peters, and M. D. Mahecha (2015), Quantifying changes in climate variability and extremes: Pitfalls and their overcoming, Geophys. Res. Lett., 42, 99909998.

Russo S, Sillmann J and Fischer E M 2015 Top ten European heat-waves since 1950 and their occurrence in the coming decades Environ. Res. Lett. 10 124003

References will be added.